# Comparison of Ki-67 Labeling Index Patterns of Diffuse Large B-Cell Lymphomas and Burkitt Lymphomas Using Image Analysis: A Multicenter Study

**DOI:** 10.3390/diagnostics11020343

**Published:** 2021-02-19

**Authors:** Yosep Chong, Tae Eun Kim, Uiju Cho, Min-Sun Jin, Kwangil Yim, Nishant Thakur, Jong Ok Kim, Inju Cho, Gyeongsin Park

**Affiliations:** 1Department of Hospital Pathology, Yeouido St. Mary’s Hospital, The Catholic University of Korea College of Medicine, Seoul 07345, Korea; ychong@catholic.ac.kr (Y.C.); nishantbiotech2014@gmail.com (N.T.); injucho81@gmail.com (I.C.); 2Department of Hospital Pathology, Uijeongbu St. Mary’s Hospital, The Catholic University of Korea College of Medicine, Uijeongbu 11765, Korea; kangse_manse@catholic.ac.kr; 3Department of Hospital Pathology, Incheon St. Mary’s Hospital, The Catholic University of Korea College of Medicine, Incheon 21431, Korea; aprstory@hanmail.net; 4Department of Hospital Pathology, St. Vincent’s Hospital, The Catholic University of Korea College of Medicine, Suwon 16247, Korea; uijucho@gmail.com; 5Department of Hospital Pathology, Bucheon St. Mary’s Hospital, The Catholic University of Korea College of Medicine, Bucheon 14647, Korea; seasy@catholic.ac.kr; 6Department of Hospital Pathology, Daejeon St. Mary’s Hospital, The Catholic University of Korea College of Medicine, Daejeon 35301, Korea; jkim@catholic.ac.kr; 7Department of Hospital Pathology, Eunpyeong St. Mary’s Hospital, The Catholic University of Korea College of Medicine, Seoul 03312, Korea; 8Department of Hospital Pathology, Seoul St. Mary’s Hospital, The Catholic University of Korea College of Medicine, Seoul 06591, Korea

**Keywords:** diffuse large B-cell lymphoma, Burkitt lymphoma, Ki-67 antigen, computer-assisted image processing, differential diagnosis, image cytometry

## Abstract

Diffuse large B-cell lymphoma (DLBCL) is the most common high-grade B-cell lymphoma found in Korea; it manifests with a variety of cellular morphologies and a high proliferation index. It is difficult to differentiate between DLBCL and Burkitt lymphoma (BL) based on immunohistochemistry, histology, and Epstein-Barr virus infection status owing to the overlap in findings. In this study, we performed comparative morphometric analysis to understand the proportional difference in Ki-67 staining between DLBCL and BL. We analyzed Ki-67-stained slides of 103 DLBCLs and 29 BLs that were pathologically confirmed using a three-tier classification system (negative, 1+, 2+, and 3+) to compare Ki-67 expression between BL and activated B-cell and germinal center B-cell subtypes of DLBCL and DLBCL with high proliferation indices (>90% of 2+ and 3+ cells). Patients with DLBCL were older than those with BL (62.1 versus 51.0 years). The number and proportion of negative cells (passenger and true negative cells) were significantly lower in BLs than those in DLBCLs (337.4, 5.9% versus 690.3, 12.4%). The number and proportion of 3+ cells were significantly higher in BLs than those in DLBCLs (5213.6, 96.3% versus 3132.4, 62.0%). BLs and DLBCLs with a high proliferation index showed similar results as those between BLs and overall DLBCLs. We were able to differentiate BLs and DLBCLs with 98.1% sensitivity and 100.0% specificity using an optimal cut-off of 97.9% of 2+/3+ Ki-67-positive cells. Thus, the Ki-67 labeling index may be a good differential biomarker for DLBCLs and BLs.

## 1. Introduction

Diffuse large B-cell lymphoma (DLBCL) is the most common high-grade B-cell lymphoma (HGBL) comprising morphologically, biologically, and clinically heterogeneous subgroups with various cellular morphologies, molecular characteristics, and a high proliferation index [1,2,3,4,5,6,7,8]. Burkitt lymphoma (BL) is a highly aggressive type of B-cell lymphoma characterized by the translocation of *MYC*, Epstein-Barr virus (EBV) infection, and Ki-67 proliferation index of ~100% [1,3,5,9]. Some types of DLBCLs morphologically resemble features of BL and have a high proliferative index of >95%. The majority of these cases possess translocations of *MYC, BCL2*, and/or *BCL6*; these are called “double hit” or “triple hit” lymphomas and sometimes exhibit a blastoid appearance [1,3,10,11,12,13]. These cases were referred to in the 2008 World Health Organization classification as B-cell lymphoma, unclassifiable, with features intermediate between DLBCL and BL, and were divided into two categories based on genetic abnormalities in the 2018 classification: HGBL with double/triple hit mutations (HGBL, DH/TH), and HGBL, not otherwise specified (Appendix A) [10,14,15].

However, this has complicated the pathologic diagnosis of HGBL since the morphologic criteria of “features intermediate between DLBCL and BL” is very arbitrary and can be interpreted subjectively by individuals, and the definition of “Near-100% Ki-67 labeling index”, characteristic for BL, has not been clearly determined yet [1,3,16]. In terms of the Ki-67 proliferation index, there can be passenger lymphocytes and residing stromal cells in the microenvironment of the lymphoma that can affect the final count of positive cells. Moreover, there is no specific cut-off value for “nearly 100% positivity in BLs” [16].

As we employed image analysis software on reporting the Ki-67 labeling index, we noticed a proportional difference in the patterns of Ki-67 staining between BLs and DLBCLs. Thus, we designed a comparative morphometric analysis and evaluated the cut-off value for the Ki-67 index in differentiating DLBCLs and BLs.

## 2. Materials and Methods

This study was approved by the Institutional Review Board of the Catholic University of Korea, College of Medicine (XC19REDI0077). Informed consent was waived according to the Institutional Review Board.

### 2.1. Samples

Hematoxylin and eosin (H&E)-stained sections and corresponding Ki-67 immunohistochemistry (IHC) slides of pathologically confirmed DLBCL and BL samples were retrieved from the archives at eight of our hospital branches at Seoul, Yeouido, Eunpyeong, Incheon, Bucheon, Uijeongbu, Daejeon St. Mary’s Hospital, and Suwon St. Vincent’s Hospital between 2013 and 2019. The H&E slides and corresponding Ki-67 IHC slides were reviewed by two hematopathology specialists (Y.C. and G.P.) in association with age, sex, molecular subtype of DLBCL (activated B-cell (ABC) and germinal center B-cell (GCB)), other IHC markers (CD3, CD20, BCL2, BCL6, CD10, c-MYC, MUM-1, etc.), EBV positivity (in situ hybridization), and fluorescence in situ hybridization (FISH) for *BCL2, BCL6*, and *MYC* translocations. The molecular subtype of DLBCL (ABC versus GCB) was reconfirmed based on Han’s algorithm, considering the original diagnosis and IHC for CD10, Bcl6, and MUM1.

### 2.2. Image Analysis

Ki-67 IHC slides were reviewed before image acquisition and cases with poor staining, such as faint staining, were excluded (1 BL and 1 DLBCL excluded). One pathologist (Y.C.) and one laboratory staff member (N.T.) acquired 4–10 representative Ki-67 images from the most proliferative areas of each case independently on two different days to access the significance of inter- and intra-observer variability on the results. The acquired images were analyzed using the United States Food and Drug Administration (US-FDA)-cleared image analysis system, GenAsisTM HiPath (Applied Spectral Imaging Ltd.,Carlsbad, CA, USA), according to the manufacturer’s instructions. Auto-white balancing was performed before capturing the image. Total and stained cells were counted using a three-tier classification system based on the staining intensity of Ki-67 estimated by the software according to the initial setting of the software (negative, 1+, 2+, and 3+; Figure 1). No pre- or post-processing was performed in any of the cases to exclude the effect of manual adjustment on the image analysis. We compared the patterns between DLBCL and BL, ABC and GCB subtypes of DLBCL, and BL and DLBCL with high proliferation indices (>90% of 2+ and 3+ cells).

### 2.3. Statistical Analysis

Comparisons of two groups were performed using the Chi-square, Fisher’s exact, Student’s *t*-test, or Mann–Whitney test. Three-group comparisons were performed using one-way analysis of variance or Kruskal–Wallis test. Statistical significance was considered at *p* < 0.05. Receiver operating characteristic (ROC) plots were constructed to calculate the area under the curve (AUC) and optimal cut-off v-alue for the Ki-67 labeling index to differentiate between DLBCLs and BLs. Statistical analysis was performed using a web-based statistical analysis, Web-R, version 4.0 (http://web-r.org) [17].

## 3. Results

### 3.1. Sample Characteristics

Table 1 summarizes the sample characteristics. Samples comprised 29 BLs and 103 DLBCLs (73 ABCs and 30 GCBs). The average age of patients with DLBCL was 11 years higher than that of patients with BL (62.1 versus 51.0 years). BL and DLBCL were found in more males than females (ratio of 1:0.61 for BLs and 1:0.69 for DLBCLs), although this difference was not statistically significant.

### 3.2. Ki-67 Labeling Index between BLs and DLBCLs

Among the roughly 5700 cells counted, the number and proportion of negative cells (passenger and true negative cells) were significantly lower in BLs than those in DLBCLs (348.8, 6.1% versus 660.7, 11.9%) (Table 1; Figure 2A–C). The number and proportion of negative, 1+, and 2+ cells were significantly lower in BLs than those in DLBCLs (337.4, 5.9% versus 690.3, 12.4% for negative, 37.8, 0.7% versus 791.4, 14.2% for 1+, and 163.8, 2.9% versus 1021.7, 18.1% for 2+ in BLs and DLBCLs, respectively). The number and proportion of positive and 3+ cells were significantly higher in BLs than those in DLBCLs (5415.1, 94.1% versus 4945.5, 87.6% for positive, 5213.6, 90.6% versus 3132.4, 55.3% for 3+ in BLs and DLBCLs, respectively). BLs comprised a smaller population of 1+ and 2+ cells and larger proportion of 3+ cells than those in DLBCLs (0.7% versus 16.6% for 1+, 3.0% versus 21.4% for 2+, 96.3% versus 62.0% for 3+ in BLs and DLBCLs, respectively). Subsequently, the proportion of 1+/2+ cells among Ki-67-positive cells was lower in BLs than that in DLBCLs (3.7% versus 38.0%), and the proportion of 2+/3+ cells among Ki-67-positive cells was higher in BLs than that in DLBCLs (99.4% versus 83.4%).

### 3.3. Ki-67 Labeling Index between ABC and GCB Subtypes of DLBCLs

There was no significant difference in the age or sex of patients with the ABC and GCB types of DLBCLs (Table 2; average age: 62.8 versus 60.5 years, male to female ratio: 1:0.87 versus 1:0.36, respectively). There was no significant difference in the number and proportion of Ki-67-negative cells (passenger and true negative cells) between the ABC and GCB types (Table 2; Figure 2D–F). The number and proportion of 2+ cells were significantly higher in the ABC subtypes than those in the GCB subtypes (1146.3, 20.4% versus 718.6, 12.4%, *p* < 0.001), while those of 3+ cells were significantly lower in ABCs than those in GCBs (2913.0, 52.5% versus 3666.2, 62.8%, *p* = 0.007/0.021). This was consistent with the proportions of 2+ or 3+ cells among the Ki-67-positive cells (24.2% versus 14.6% for 2+ cells, *p* < 0.001, 57.7% versus 72.5% for 3+ cells, *p* = 0.001). However, although the number and proportion of 1+ cells were significantly higher in ABCs than those in GCBs (871.7, 15.6% versus 595.8, 10.9%, *p* = 0.036/0.046), the proportion of 1+ cells among the Ki-67-positive cells was not significantly different between the two types (18.1% in ABCs versus 12.9% in GCBs, *p* = 0.063). Subsequently, the proportions of 1+/2+ and 3+ cells among the positive cells were significantly higher and lower, respectively, in ABCs than those in GCBs. The proportion of total positive cells was not significantly different between the two types. Similarly, the proportion of 1+ and 2+/3+ cells among the positive cells was not significantly different between the two types.

### 3.4. Ki-67 Labeling Index between BLs and ABC and GCB Subtypes of DLBCLs

The number and proportion of negative, 1+, 2+, 3+, 1+/2+, and 2+/3+ cells were significantly different between BLs, ABCs, and GCBs (*p* = 0.013; Appendix A). The number and proportion of negative cells (passenger and true negative cells) were highest in GCBs (772.6, 13.9%) followed by ABCs (656.5, 11.7%) and lowest in BLs (337.4, 5.9%). The number and proportion of 1+ and 2+ cells were highest in ABCs (871.7, 15.6% and 1146.3, 20.4%) followed by GCBs (595.8, 10.9% and 718.6, 12.4%) and lowest in BLs (37.8, 0.7% and 163.8, 2.9%). The number and proportion of 3+ cells were highest in BLs (5213.6, 90.6%) followed by GCBs (3666.2, 62.8) and lowest in ABCs (2913.0, 52.2%). The proportion of 1+, 2+, and 3+ cells among Ki-67-positive cells was consistent with the aforementioned findings. The proportion of 1+/2+ among positive cells was significantly higher in ABCs (42.3%) followed by GCBs (27.5%) and lowest in BLs (3.7%). On the other hand, the proportion of 2+/3+ among the positive cells was significantly higher in BLs (99.3%) followed by GCBs (87.1%) and lowest in ABCs (81.9%). Collectively, BLs tended to exclusively have 3+ cells and GCBs tended to have more 2+/3+ cells, particularly more 3+ cells, and slightly more passenger and true negative cells than those in the ABCs.

### 3.5. Ki-67 Labeling Index between BLs and DLBCLs with a High Proliferation Index

To simulate a clinical setting with a high Ki-67 labeling index, we arbitrarily defined the highly proliferating tumors as those with >90% of 2+/3+ Ki-67-positive cells. All the 29 BL cases and 42 of the DLBCL cases (23 ABCs and 19 GCBs) met our criteria and were included in the comparison (Table 3). The proportion of cells was almost identical to those in BLs and total DLBCLs (Table 1 and Table 3). Although DLBCLs with a high proliferation index showed an average of 94.3% of 2+/3+ cells among positive cells similar to that of BLs (99.3%), the number and proportion of 3+ cells were higher in BLs than those in DLBCLs (96.3% versus 76.9%, *p* < 0.001) and those of negative, 1+, and 2+ cells were higher in DLBCLs than those in BLs (11.7%, 5.7%, 17.4% versus 5.9%, 0.7%, and 3.0% for negative, 1+, and 2+ cells, respectively; *p* = 0.011/*p* < 0.001).

### 3.6. ROC Curve, AUC, and Optimal Cut-Off Value for Differentiating BLs and DLBCLs

Using the exact numbers of positive cells based on Ki-67 staining intensity that were significantly different between BLs and DLBCLs, we performed an ROC analysis to detect the most effective value for differentiating BLs and DLBCLs (Figure 3). The AUC was highest for the proportion of 2+/3+ Ki-67-positive cells (0.989) with 98.1% sensitivity and 100.0% specificity, which was also the same with the proportion of 1+ Ki-67-positive cells. The optimal cut-off value was 97.9% for the proportion of 2+/3+ cells among the Ki-67-positive cells (2.1% for the proportion of 1+ cells among the Ki-67-positive cells) (Figure 4). Thus, samples with >97.9% of 2+/3+ and <2.1% 1+ Ki-67-positive cells can be considered BL rather than DLBCL.

### 3.7. Inter- and Intra-Observer Variability (Comparison between the Samples Reviewed by Different People and on Different Days)

Within the DLBCL and BL groups, the Ki-67 labeling proportion of the samples reviewed by the pathologist (Y.C.) and the laboratory staff member (N.T.) were compared. Sixty-five cases including 15 BL cases were reviewed by Y.C. and sixty-seven cases including 14 BL cases were reviewed by N.T. Thirty-five cases including 8 BL cases were reviewed by Y.C. at first and the rest of the cases were reviewed after a week. Thirty-five cases including 7 BL cases were reviewed by N.T. at first and the rest of the cases were reviewed after a week. There were no significant differences in any of proportional groups between the two sample groups reviewed by Y.C. and N.T. (data not shown) either in DLBCLs and BLs. There were no significant differences in any of the proportional groups between the two sample groups reviewed on two different days in DLBCLs and BLs and by Y.C. and N.T. (data not shown).

## 4. Discussion

In this study, we first found that BL samples had a lower number/proportion of negative and 1+ cells and a higher number/proportion of 2+/3+ cells, particularly 3+ cells, than those of DLBCLs, regardless of the subtypes. That is, in BLs, the tumor cells were a more homogeneous population with intense nuclear Ki-67 staining and with a reduced proportion of passenger cells in the background. Second, using a cut-off value of 97.9% for the 2+/3+ cells among the Ki-67 positive cells, it was possible to differentiate between BLs and DLBCLs by image analysis with very high sensitivity and specificity. Third, the GCB subtype of DLBCLs possessed a higher proportion of 2+/3+ cells, especially 3+ cells, than those in the ABC subtype, which mimicked the Ki-67 staining pattern found for BLs.

Consistent with the histological and molecular features of BL and DLBCL, we observed distinctive IHC staining patterns of Ki-67 in BL and DLBCL samples. BL samples tend to have a very homogenously monotonous morphology with indirect evidence of high proliferation, such as squared-off nuclear borders, frequent mitosis, and apoptosis [1,3,10,16]. The molecular hallmark of BL is the translocation of *MYC*, almost exclusively [1,3,18]. In contrast, DLBCL samples have a wide spectrum of histological features with various genomic aberrations that represent a more complex pathogenesis [1,3,19]. Considering the complex nature of molecular changes of these diseases, the difference in Ki-67 staining patterns cannot be explained with simple theory. However, one thing that is clear based on our results is that the strong and homogenous positivity of Ki-67 in BL can be objectively measured and effectively used as a good marker to help distinguish BL from DLBCL.

The Ki-67 labeling index, one of the most widely used markers of proliferation in oncology, is often estimated simply by counting positively stained cells among all cells [20,21,22]. In a recent study, however, it turned out that the Ki-67 labeling index is a graded rather than a binary marker of proliferation versus quiescence [20]. In that study, the authors found that Ki-67 levels are hard-wired into cell-cycle progression and exit and are graded continuously in G0 and G1 and accumulate from S to M phases [20]. According to this finding, the high proportion of 2+/3+ cells in BLs found in our study represents the higher proportion of cells in S and M phases and the rapid cell cycle of BLs.

In this study, we evaluated the “nearly 100% Ki-67 labeling index in BLs”. In a recent study that suggested algorithmic diagnosis of BLs using Ki-67, the authors suggested to use a scoring system for certain morphologic, immunohistochemical, and molecular findings to differentiate BLs from other HGBLs [16]. They suggested scoring 2 for Ki-67 ≥ 95 and 1 for Ki-67 of 90–95% [16]. However, there is no definite rationale for how those figures were made [16]. In lymphoid neoplasms, objective morphometric assessment of certain IHC markers is a very challenging task, even for hematopathology specialists [16,23]. Owing to the basic oncogenic nature of lymphoid neoplasms, lymphoma cells and residing reactive lymphoid cells, such as T-cells, are frequently admixed within the tumor. In extranodal lymphomas, such as BL and DLBCL, residing stromal cells, blood vessels, and benign epithelial cells are also often encountered as passenger cells [1,3]. Thus, accurate estimation of the true population of Ki-67-positive cells among all tumor cells is theoretically impossible in Ki-67 IHC interpretation. For this reason, morphometric analysis using image analysis software is being introduced in IHC analysis [23,24]. This technique is widely used for the assessment of estrogen receptors, progesterone receptors, and c-erbB2 in breast cancer [23,24,25,26]. It is also useful for estimating the intensity of positive cells more objectively than with human eyes. In our study, we compared the number of negative and positive cells and the proportion of 1+, 2+, and 3+ cells among the total cells and all positive cells between the two groups and the subgroups of DLBCLs. As a result, we found a lower number of passenger lymphocytes/residing stromal cells (or true negative tumor cells) and a higher proportion of 3+ cells among the Ki-67-positive cells in BLs than those in DLBCLs. Moreover, we observed that the most sensitive and specific method of differentiating BL and DLBCL is to compare the proportion of 2+/3+ cells among positive cells with a cut-off of 97.9%, which is very simple and straightforward.

Furthermore, we characterized the Ki-67 expression pattern in the ABC and GCB subtypes of DLBCL. Interestingly, GCBs showed a higher proportion of 2+/3+ cells, especially 3+ cells, than those in the ABCs, which was similar to the population observed in BLs. This can be explained by the fact that both BLs and the GCB subtype of DLBCLs are considered to be originated from germinal center B-cells while the ABC subtype of DLBCLs are originated from early plasmablastic or post-germinal center cells [7].

The multicenter nature of this study helps highlight our data. Samples were collected from eight hospitals covering most of the metropolitan area of Seoul, with almost one-fourth of the total population of South Korea. However, the limitation of this study is the sole involvement of the Korean population.

In addition, the comparison on the inter- and intra-observer variability showed that there is minimal or no influence of the person performing the review or time on the image analysis, and image analysis can provide very reproducible results, minimizing the human error, which is the major strength of digital image analysis over manual morphometric analysis [23].

We also tried to compare BL and DLBCL with *MYC* translocation, or double hit/triple hit translocation during the study design [12,27]. Unfortunately, not all FISH results of the enrolled DLBCLs in this study were available. Further study with more DLBCLs with full FISH results is desirable in the near future.

## 5. Conclusions

Taken together, we performed image analysis and found that the Ki-67 labeling index might serve as a very sensitive and specific marker for differentiating between BL and DLBCL among HGBL with a high proliferation index. Combining this Ki-67 labeling pattern to histology, IHC, and EBV-specific assays may help reach a more conclusive pathological diagnosis of the cancer and help with the choice of adequate molecular tests for potential double hit/triple hit lymphomas.

## Figures and Tables

**Figure 1 diagnostics-11-00343-f001:**
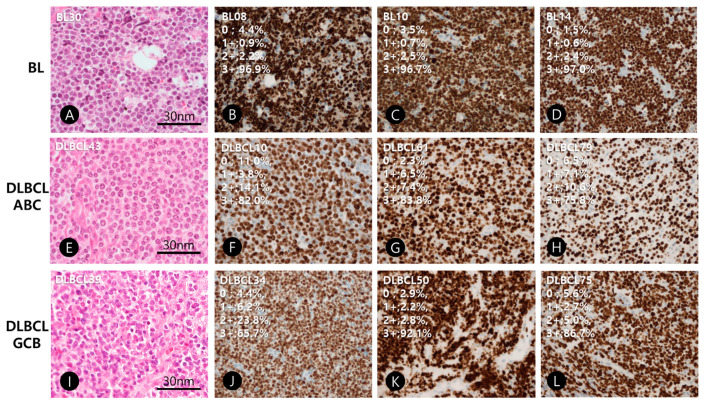
Representative figures of hematoxylin and eosin stains and Ki-67 immunohistochemistry of Burkitt lymphoma (BL), diffuse large B-cell lymphoma, activated B-cell type (DLBCL, ABC) and germinal center cell type (DLBCL, GCB). (**A**–**D**); BLs (Case No. BL30, BL08, BL10, and BL14), (**E**–**H**); DLBCL, ABC (Case No. D043, D010, D061, D079), (**I**–**L**); DLBCL, GCB (Case No. D039, D034, D050, D075).

**Figure 2 diagnostics-11-00343-f002:**
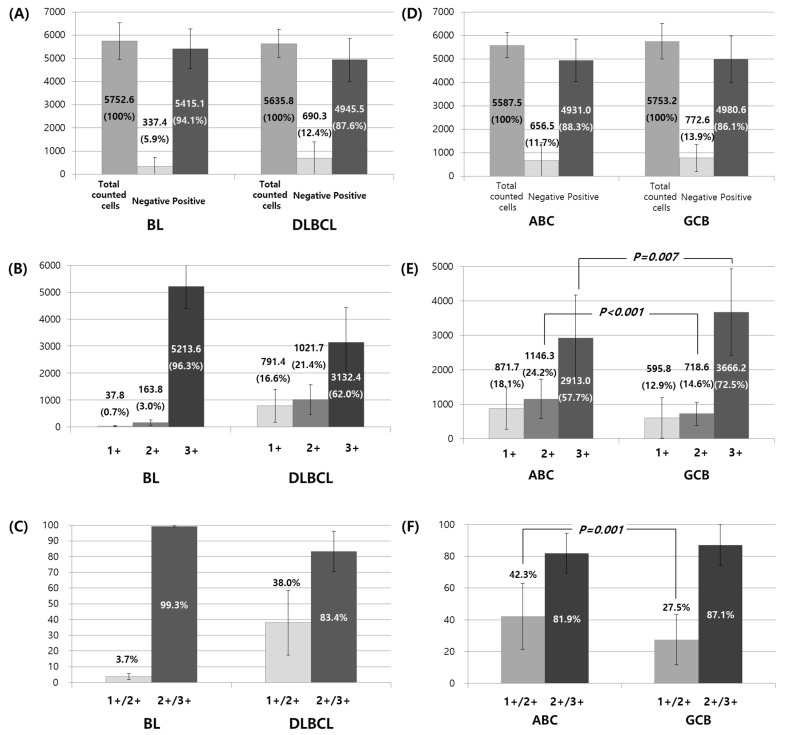
Comparison of Ki-67 labeling index patterns between Burkitt lymphomas (BLs) and diffuse large B-cell lymphomas (DLBCLs) (**A**–**C**) and between activated B-cell (ABC) and germinal center cell (GCB) subtypes of DLBCLs (**D**–**F**). Chi-square, Fisher’s exact, Student’s *t*-test, or Mann–Whitney test were used for two-group comparisons. The one-way analysis of variance or Kruskal–Wallis test were used for three-group comparisons.

**Figure 3 diagnostics-11-00343-f003:**
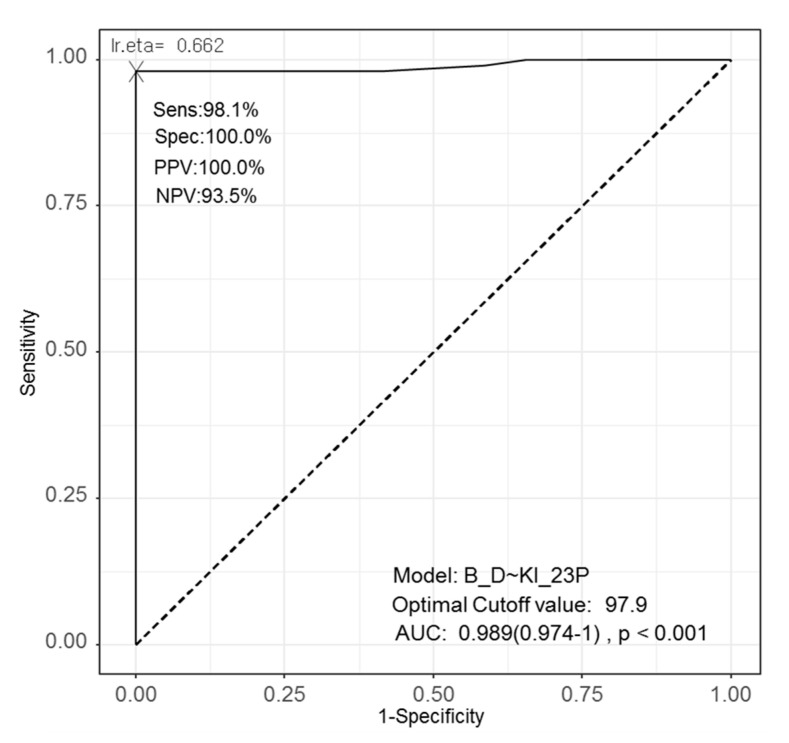
Receiver operating characteristic (ROC) curve using 2+ and 3+ cell proportions among positive cells, area under the curve (AUC), and the optimal cut-off value for differentiating Burkitt lymphomas (BLs) and diffuse large B-cell lymphomas (DLBCLs). Using the cut off value of 97.9%, the sensitivity is 98.1% and the specificity is 100.0%.

**Figure 4 diagnostics-11-00343-f004:**
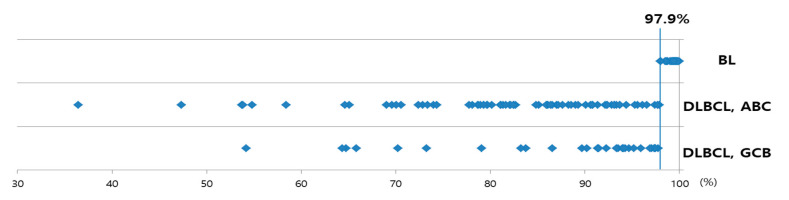
Dot plot of 2+/3+ cell proportions among positive cells of Burkitt lymphomas (BLs), diffuse large B-cell lymphomas (DLBCLs), and activated B-cell (ABC) and germinal center cell (GCB) subtypes.

**Table 1 diagnostics-11-00343-t001:** Comparison of Ki-67 labeling index patterns between Burkitt lymphomas (BLs) and diffuse large B-cell lymphomas (DLBCLs).

Clinicopathologic Parameters	Units	BLs		DLBCLs		*p*-Value
		(*n* = 29)		(*n* = 103)		
Age	years	51.0 ± 24.8	range 5–81	62.1 ± 14.6	range 17–89	***0.027***
Sex						0.951
Male	No.	18 (62.1%)		61 (59.2%)		
Female	No.	11 (37.9%)		42 (40.8%)		
M:F		1:0.61		1:0.69		
Total counted cells	No.	5752.6 ± 786.2	4269–7613	5635.8 ± 612.8	3437–8239	0.397
Negative cells	No.	337.4 ± 385.4	23–1468	690.3 ± 715.2	7–2882	***0.001***
	%	5.9 ± 6.6	0.4–26.6	12.4 ± 13.2	0.1–53.5	***<0.001***
Positive cells	No.	5415.1 ± 850.9	4002–7442	4945.5 ± 928.1	2168–7421	***0.016***
	%	94.1 ± 6.6	73.4–99.6	87.6 ± 13.2	46.5–99.9	***<0.001***
1+	No.	37.8 ± 24.9	2–103	791.4 ± 608.1	29–3337	***<0.001***
	%	0.7 ± 0.5	0.0–2.0	14.2 ± 11.1	0.4–58.7	***<0.001***
2+	No.	163.8 ± 102.3	10–493	1021.7 ± 547.2	148–2451	***<0.001***
	%	2.9 ± 1.7	0.2–9.2	18.1 ± 9.5	2.8–45.5	***<0.001***
3+	No.	5213.6 ± 825.7	3737–7269	3132.4 ± 1296.8	147–6422	***<0.001***
	%	90.6 ± 6.2	67.8–95.5	55.3 ± 21.3	2.6–92.1	***<0.001***
Proportion of 1+ in positive cells	%	0.7 ± 0.5		16.6 ± 12.8		***<0.001***
Proportion of 2+ in positive cells	%	3.0 ± 1.8		21.4 ± 12.3		***<0.001***
Proportion of 3+ in positive cells	%	96.3 ± 2.0		62.0 ± 20.6		***<0.001***
Proportion of 1+/2+ in positive cells	%	3.7 ± 2.0		38.0 ± 20.6		***<0.001***
Proportion of 2+/3+ in positive cells	%	99.3 ± 0.5		83.4 ± 12.8		***<0.001***

Significant *p*-values are bolded and italicized.

**Table 2 diagnostics-11-00343-t002:** Comparison of Ki-67 labeling index patterns between activated B-cell (ABC) and germinal center cell (GCB) subtypes of diffuse large B-cell lymphomas (DLBCLs). Significant *p*-values are bolded and italicized.

Clinicopathologic Parameters	Units	ABC Type		GCB Type		*p*-Value
		(*n* = 73)		(*n* = 30)		
Age	years	62.8 ± 14.5	range 24–89	60.5 ± 15.2	range 17–84	0.475
Sex						0.099
Male	No.	39 (53.4%)		22 (73.3%)		
Female	No.	34 (46.6%)		8 (26.7%)		
M:F		1:0.87		1:0.36		
Total counted cells	No.	5587.5 ± 541.1	4520–8239	5753.2 ± 757.6	3437–7133	0.282
Negative cells	No.	656.5 ± 767.6	7–2882	772.6 ± 571.4	13–2438	0.457
	%	11.7 ± 14.0	0.1–53.5	13.9 ± 11.1	0.2–46.4	0.457
Positive cells	No.	4931.0 ± 905.0	2378–7421	4980.6 ± 997.1	2168–6866	0.807
	%	88.3 ± 14.0	46.5–99.9	86.1 ± 11.1	53.6–99.8	0.457
1+	No.	871.7 ± 599.3	115–3337	595.8 ± 594.3	115–2344	0.036
	%	15.6 ± 10.8	2.0–58.7	10.8 ± 10.8	2.1–44.8	0.046
2+	No.	1146.3 ± 570.4	335–2451	718.6 ± 333.6	148–1520	***<0.001***
	%	20.4 ± 9.9	5.6–45.5	12.4 ± 5.4	2.8–24.5	***<0.001***
3+	No.	2913.0 ± 1256.9	147–5253	3666.2 ± 1227.8	1422–6287	***0.007***
	%	52.2 ± 22.2	2.6–88.5	62.9 ± 16.8	27.0–92.1	***0.021***
Proportion of 1+ in positive cells	%	18.1 ± 12.6		12.8 ± 12.3		0.063
Proportion of 2+ in positive cells	%	24.2 ± 13.1		14.6 ± 5.9		***<0.001***
Proportion of 3+ in positive cells	%	57.7 ± 20.8		72.6 ± 15.4		***0.001***
Proportion of 1+/2+ in positive cells	%	42.3 ± 20.8		27.4 ± 15.4		***0.001***
Proportion of 2+/3+ in positive cells	%	81.9 ± 12.6		87.2 ± 12.3		0.063

**Table 3 diagnostics-11-00343-t003:** Comparison of Ki-67 labeling index between Burkitt lymphomas (BLs) and diffuse large B-cell lymphomas (DLBCLs) with a high proliferation index (2+ and 3+ cells >90%). Significant *p*-values are bolded and italicized.

Clinicopathologic Parameters	Units	BLs		DLBCLs with a High Proliferation Index (2+/3+ > 90%)		*p*-Value
		(*n* = 29)		(*n* = 42, 23 ABC, 19 GCB)		
Age	years	51.0 ± 24.8	range 5–81	62.2 ± 14.9	range 25–85	***0.034***
Sex						1.000
Male	No.	18 (62.1%)		26 (61.9%)		
Female	No.	11 (37.9%)		16 (38.1%)		
M:F		1:0.61		1:0.62		
Total counted cells	No.	5752.6 ± 786.2	4269–7613	5759.9 ± 575.9	5004–7133	0.964
Negative cells	No.	337.4 ± 385.4	23–1468	669.2 ± 681.5	7–2737	***0.011***
	%	5.9 ± 6.6	0.4–26.6	11.7 ± 12.6	0.1–53.5	***0.013***
Positive cells	No.	5415.1 ± 850.9	4002–7442	5090.7 ± 899.6	2378–6866	0.131
	%	94.1 ± 6.6	73.4–99.6	88.3 ± 12.6	46.5–99.9	***0.013***
1+	No.	37.8 ± 24.9	2–103	281.6 ± 126.2	29–519	***<0.001***
	%	0.7 ± 0.5	0.0–2.0	5.0 ± 2.3	0.4–9.3	***<0.001***
2+	No.	163.8 ± 102.3	10–493	835.6 ± 503.5	148–2400	***<0.001***
	%	2.9 ± 1.7	0.2–9.2	14.6 ± 8.9	2.8–45.5	***<0.001***
3+	No.	5213.6 ± 825.7	3737–7269	3973.4 ± 1095.3	898–6422	***<0.001***
	%	90.6 ± 6.2	67.8–95.5	68.7 ± 17.1	17.0–92.1	***<0.001***
Proportion of 1+ in positive cells	%	0.7 ± 0.5		5.7 ± 2.6		***<0.001***
Proportion of 2+ in positive cells	%	3.0 ± 1.8		17.4 ± 12.6		***<0.001***
Proportion of 3+ in positive cells	%	96.3 ± 2.0		76.9 ± 13.7		***<0.001***
Proportion of 1+/2+ in positive cells	%	3.7 ± 2.0		23.1 ± 13.7		***<0.001***
Proportion of 2+/3+ in positive cells	%	99.3 ± 0.5		94.3 ± 2.6		***<0.001***

## Data Availability

The data presented in this study are openly available in Re-searchgate.net at doi:10.13140/RG.2.2.30369.84328.

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
