# Peer review of "Comparison of Ki-67 Labeling Index Patterns of Diffuse Large B-Cell Lymphomas and Burkitt Lymphomas Using Image Analysis: A Multicenter Study"

_diagnostics, 2021, doi:10.3390/diagnostics11020343_

Round 1

Reviewer 1 Report

The authors performed a morphometric analysis of ki-67 staining in BL and DLBCL and concluded that ki-67 labeling index can serve as a sensitive and specific marker for differentiation of the two entities. I have a few comments and suggestions.

  1. One of the diagnostic challenges is to differentiate BL from DLBCL with c-myc translocation (not including HGBL with double-hit or triple-hit) as 5% to 15% of DLBCLs carry c-myc translocation.  I think that it would be better to perform a comparative analysis of ki-67 labeling on the two groups.
  2. What percent of cases were excluded due to faint staining?
  3. Please add representative HE pictures for BL and DLBCL in figure 1.

Author Response

The authors performed a morphometric analysis of Ki-67 staining in BL and DLBCL and concluded that ki-67 labeling index can serve as a sensitive and specific marker for differentiation of the two entities. I have a few comments and suggestions.

  1. One of the diagnostic challenges is to differentiate BL from DLBCL with c-myc translocation (not including HGBL with double-hit or triple-hit) as 5% to 15% of DLBCLs carry c-myc translocation. I think that it would be better to perform a comparative analysis of ki-67 labeling on the two groups.

    Answer: Thank you for your valuable suggestion. We also thought it is a very good idea and also tried to compare BLs and DLBCL with c-myc translocation groups when we first designed this study. However, not all of the enrolled cases were examined for c-myc translocation by FISH. And we thought we would not be able to find enough number of DLBLCs with c-myc translocation that can be statistically meaningful even if we perform FISH on every case because each group should be more than 30 to draw out meaningful data but c-myc+ DLBCL will be less than 10 cases. So we would not be able to do this analysis and maybe we might need to gather some more cases for further analysis in future. We added this in the discussion part as below.

We also tried to compare BL and DLBCL with MYC translocation, or double hit/triple hit translocation during the study design. Unfortunately, however, not all FISH results of the enrolled DLBCLs in this study were available. Further study with more DLBCLs with full FISH results can be suggested desirable in near future.

  1. What percent of cases were excluded due to faint staining?

    Answer: Thank you for your question. Two BLs and two DLBCLs were excluded due to tiny amount of material and faint staining, which is 3% out of all cases. We added this in the results part of the manuscript.

  2. Please add representative HE pictures for BL and DLBCL in figure 1.

Answer: Thank you for your suggestion. We added representative HE figures for BL and DLBCLs as per reviewer’s suggestion.

Reviewer 2 Report

In figure 2 please explain how many replicates were done and the statistical analysis conducted in the figure legend.

You need much more references. For a good paper at least 25 references are needed. Try to reach 30 in both discussions and introduction. For example in introduction you can explain the incidence of the disease and cite Siegel et al 2021 Cancer Statistics.

Remove section 6 if there are no patents for this specific subject.

Author Response

  1. In figure 2 please explain how many replicates were done and the statistical analysis conducted in the figure legend.

 Answer: Thank you for your suggestion. According to replicates, please note that this study does not include any laboratory experiments thus there is no replicates but we only took the representative 4-10 Ki-67 IHC images per each case and analyzed the average counts of positive cells according to intensity and statistically compared the average counts/percentage. So mentioning about replicates is a bit pointless here. However, we added statistical methods used in this analysis in the figure legend as per your request.

  1. You need much more references. For a good paper at least 25 references are needed. Try to reach 30 in both discussions and introduction. For example in introduction you can explain the incidence of the disease and cite Siegel et al 2021 Cancer Statistics.

 Answer: Thank you for your suggestion. We added more references including Seigel et al. 2021.

  1. Remove section 6 if there are no patents for this specific subject.

Answer: Thank you for your careful observation. We have removed the section.
